# Measurements of the water balance components of a large green roof in Greater Paris Area

Pierre-Antoine Versini[1], Filip Stanic[1,2], Auguste Gires[1], Daniel Scherzer[1], Ioulia Tchiguirinskaia[1]

[1]Hydrology, Meteorology and Complexity, Ecole des Ponts ParisTech, Champs-sur-Marne, 77455, France
[2]Navier, Ecole des Ponts ParisTech, Champs-sur-Marne, 77455, France

*Correspondence to*: P.-A. Versini (pierre-antoine.versini@enpc.fr)

**Abstract.** The Blue Green Wave of Champs-sur-Marne (France) represents the largest green roof (1 ha) of the Greater Paris Area. The Hydrology, Meteorology and Complexity lab of Ecole des Ponts ParisTech has chosen to convert this architectural building as a full-scale monitoring site devoted to studying the performance of green infrastructures in stormwater management. For this purpose, the relevant components of the water balance during a rainfall event have been monitored: rainfall, water content in the substrate and the discharge flowing out of the infrastructure. Data provided by adapted measurement sensors were collected during 78 days between February and May 2018. The related raw data and a python program transforming them into hydrological quantities and providing some preliminary elements of analysis have been made available. These measurements are useful to better understand the hydrological processes (infiltration and retention) conducting green roof performance, and their spatial variability due to substrate heterogeneity.

Link to the data set (Versini et al., 2019): https://doi.org/10.5281/zenodo.3687775
(doi: 10.5281/zenodo.3687775)

Keywords: green roof; stormwater management; water balance

## 1 Introduction

Considered as Blue Green Solutions (BGS), green roofs are recognized as multifunctional assets able to provide several ecosystem performances (Francis and Jensen, 2017; Oberndorfer et al., 2007) to face climate change and unsustainable urbanization consequences (as biodiversity conservation or thermal insulation). They appear to be particularly relevant in stormwater management as they have the ability to store a more or less significant portion of precipitation (Stovin et al., 2012; Versini et al., 2016). Indeed, at the building scale, green roofs contribute to: (i) reduce runoff volume at the annual scale, and (ii) attenuate and delay the peak at the rainfall event scale. This performance depends on the green roof properties (substrate depth, porosity, or vegetation type), rainfall intensity and antecedent soil moisture conditions (Berndtsson, 2010). Considered as stormwater Source Control facilities, they can act to manage rainwater at a small-scale (about $10^2 - 10^3 \text{ m}^2$) to solve or prevent intermediate scale ($10^4 - 10^6 \text{ m}^2$) stormwater issues.

By increasing the storage of water, green roofs contribute to reduce the rainwater reaching the stormwater management network. It is particularly relevant to comply with regulation rules that are generally adopted by local authorities in charge of stormwater management, usually divided in two categories: flow-rate based regulation and volume-based regulations (Petrucci et al., 2013). As green roofs perform in both retention (ability to permanently hold back water by storing the water for

subsequent removal by evapotranspiration) and detention (ability to temporarily hold back the water) (Johannessen et al., 2018), they can be used as relevant tools to ensure both kinds of regulation.

Indeed, for a green roof located in the Greater Paris Area (characterized by a temperate climate), the water balance during a rainfall event can be reduced to 3 components (see Eq. 1) as evapotranspiration can be neglected:

$$P = Q + \Delta S \tag{1}$$

Where $P$ is the precipitation, $Q$ the discharge flowing out of the structure, and $\Delta S$ the variation of water stored in the substrate conducting both retention and detention properties. All quantities are expressed in $m^3$.

Many experimental set-ups were implemented to monitor, assess and understand the hydrological behavior of green roofs (see (Berndtsson, 2010) for a review). Most of them were conducted on small green roof modules or plots (Berretta et al., 2014; Getter et al., 2007; Li and Babcock, 2015; Locatelli et al., 2014; Loiola et al., 2019; Poë et al., 2015; Stovin et al., 2015; Wong and Jim, 2015; Zhang et al., 2015) characterized by an area ranging 0.5 to 3 $m^2$. These modular structures make possible the modification of green roof configuration and study of the effects of substrate (depth and nature), vegetation type, slope, or climate conditions on its performance. Some of them were also monitored in controlled conditions (Ouldboukhitine et al., 2011; Poë et al., 2015) to assess the respective impacts of temperature, irrigation, and light on green roof behavior for instance.

In addition, few studies were conducted at full-scale green roofs. Indeed, such large infrastructures were usually not planned to be monitored during their construction, and became hard to be monitored after For instance, once built, electric connection is rarely compatible with the conservation of the roof sealing. To the knowledge of the authors, only the following works can be mentioned.

(Palla et al., 2009a) studied an instrumented portion (170 $m^2$) of a green roof in Genoa (Italy) under Mediterranean climate. This pilot site was equipped to monitor the different components of the water balance with: a meteorological station for rainfall, several Time Domain Reflectometry probes installed horizontally along a vertical profile for retention in the substrate, and a triangular weir and a tipping bucket devices to follow the outflowing discharge.

(Hakimdavar et al., 2016) used the data collected on three full-scale extensive green roofs in New York City (USA) to validate a modeling approach based on the Soil Water Apportioning Method (SWAM). Under a humid continental climate, these monitored drainage areas ranged between 310 and 940 $m^2$. The three main components of the water balance were measured: rainfall with a weather station, water content with soil moisture and water content reflectometer sensors, and discharge with a custom designed weir placed in the drain of the green roof.

(Fassman-Beck et al., 2013) assessed several green roofs in Auckland (New Zeland) under sub-tropical climate. Their areas ranged between 17 and 171 $m^2$. As the experimental setup was focused on rainfall-runoff relationship, only these components were measured: rainfall with a tipping bucket rain gauge and discharge (deduced from water level) from a water pressure transducer and a custom-designed orifice restricted device.

(Cipolla et al., 2016) analyzed runoff from a 60 $m^2$ extent green roof in Bologna (Italy) characterized by a humid temperate sub-continental climate. Continuous weather data and runoff were especially monitored for modeling development. Runoff was estimated by using an in-pipe flow meters consisting of a runoff chamber with an outlet weir and an ultrasonic sensor (to detect water level). The site was also equipped with a weather station measuring several meteorological variables (rainfall, wind speed, wind direction, relative humidity, atmospheric temperature, …).

Although these works were focused on the hydrological behavior of green roofs, few of them have actually monitored the 3 components of the water balance. Rainfall and discharge were generally considered as sufficient to assess its performance. Some additional studies can also be mentioned, but as they were focused on other topics (evapotranspiration processes (Feng et al., 2018), or water quality (Buffam et al., 2016)), only one component on the water balance was assessed.

The full-scale monitoring experiments mentioned above also suffered from two limitations. First, they were still dedicated to rather small green roof areas. As the hydrological performance of a green roof is influenced by the size of the plot (water detention depends on water routing in the structure for instance), larger infrastructure should be studied. Second, very few measurements are performed (usually only one!) to assess water content on the whole vegetated surface. Indeed, green roof substrates –which are usually largely composed of mineral components – are very heterogeneous, causing variability in their infiltration and retention capacities. Therefore, large-scale monitoring set-ups able to capture this heterogeneity are required to better understand green roof hydrological behavior and to study the space-time variability of the involved processes.

Based on these considerations, this paper aims to present and make available the water balance data collected on a large green roof (called Blue Green Wave) located close to Paris (temperate climate) in order to study its hydrological behavior and its ability to be used as stormwater management tool. The monitoring set-up has been specifically tailored to take into account the space-time variability of the water balance components.

## 2 Materials and method

### 2.1 The Blue Green Wave

The Blue Green Wave (BGW) is a large (1 ha) wavy-form vegetated roof located in front of Ecole des Ponts ParisTech (Champs-sur-Marne, France). For now it represents the largest green roof of the Greater Paris area. From its implementation in 2013, the BGW has been considered as a demonstrative site oriented to Blue Green Solutions research (Versini et al., 2018). This experimental set-up started during the European Blue Green Dream (BGD) project (http://bgd.org.uk/, funded by Climate-KIC) that aimed to promote a change of paradigm for efficient planning and management of new or retrofitted urban developments by promoting the implementation of BGS (Maksimovic et al., 2013). The monitoring was anticipated and the building was adapted to experimental purpose during its construction. It was also been supported by RadX@IdF, a regional project that aimed at analysing the benefits of high-resolution rainfall measurement for urban storm water management. Today the BGW is also part of the Fresnel multi-scale observation and modelling platform created in the Co-Innovation Lab at École des Ponts ParisTech. Fresnel aims to facilitate synergies between research and innovation, as well as the pursuit of theoretical research, the development of a network of international collaborations, and various aspects of data science (https://hmco.enpc.fr/portfolio-archive/fresnel-platform/).

From a technical point of view, the BGW is covered by two types of vegetation: green grass that represents the large majority of its area and a mix of perennial planting, grasses and iris bulbous (see Figure 1). Vegetation is laid on a substrate layer of about 200 mm depth (SOPRAFLOR I966), a filter layer made of synthetic fiber (SOPRATEX 650), and a drainage layer made of expanded polystyrene (SOPRADRAIN). The vertical profile of the structure is presented in Figure 2. The substrate was initially composed of volcanic soil (around 85%) completed by organic matter. It is worth noting that 50 % of the grains (in mass) are larger than 1.6 mm and 13 % of fine particles are smaller than 80 μm. The main physical properties of the substrate are synthetized in Table 1 (see Stanic et al., 2019 for a detailed description including grain size distribution, water retention and hydraulic conductivity curves).

| Initial composition of the substrate | Porosity | Dry Density | Saturated hydraulic conductivity |
|---|---|---|---|
| 85% of mineral matters and 15% or organic matters | 40% | 1442 g/l | $8.11 \times 10^{-6}$ m/s |

**Table 1. Physical properties of the BGW substrate**

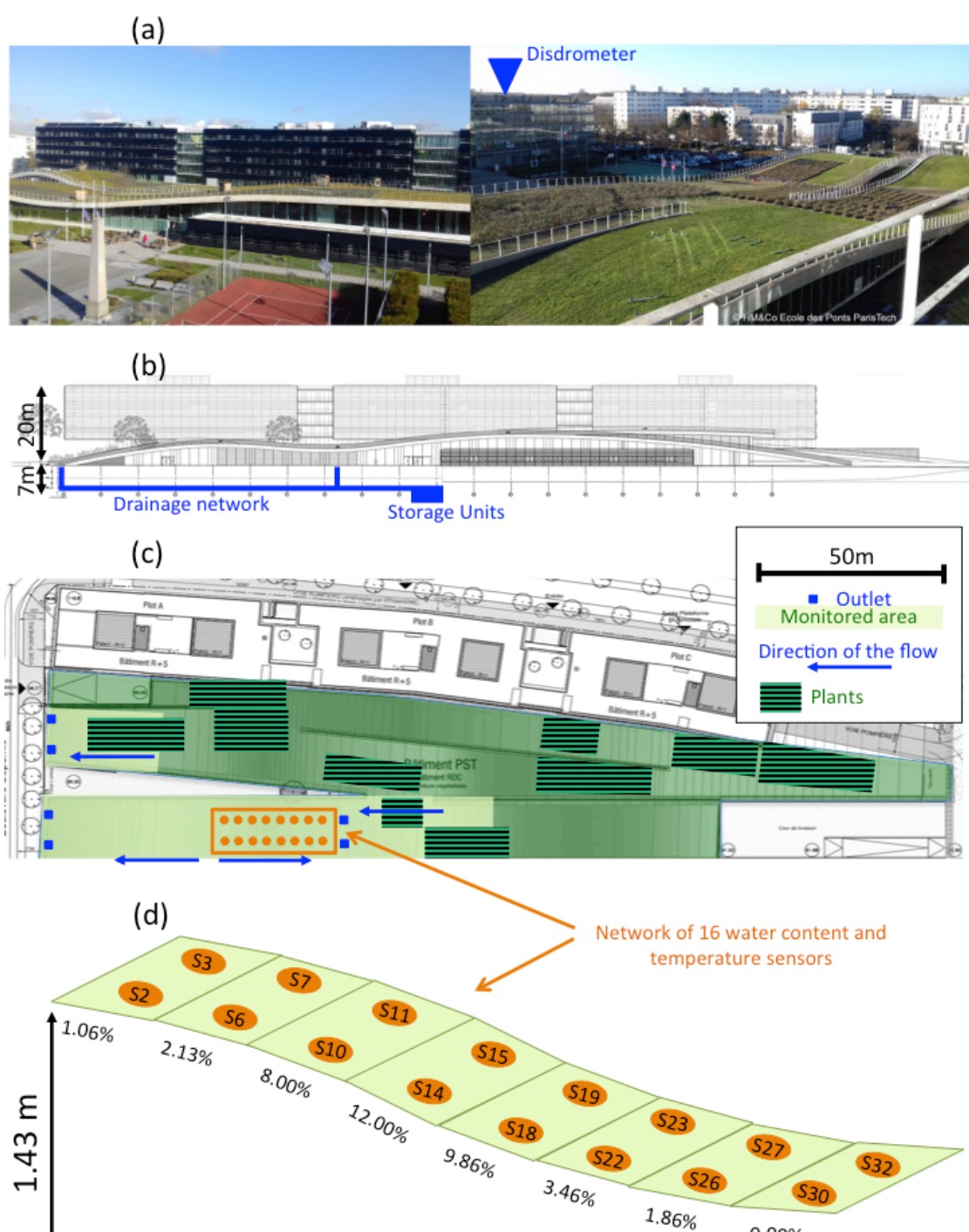

**Figure 1. The Blue Green Wave monitoring site of ENPC: (a) pictures, (b) vertical representation and flow path lengths, (c) aerial representation showing the monitored area, (d) profile of the section where the water content sensors were implemented indicating the slopes**

From a hydrological point of view, the BGW is connected to three storage units that collect rainwater coming from the roof (with pipes) but also from several impervious parts around the greened building. One of the storage units is preceded by a smaller unit dedicated to irrigation. The water is then routed to a large retention basin to collect excess volumes of water during a rainfall event before being routed to the stormwater management network. This retention basin was designed (and oversized) because it was considered that the green roof (representing 50% of the total contributive area) was totally impervious without any retention capacity. Until now in France, there is neither rule nor guideline devoted to retention basin sizing that takes into account the retention properties of green areas.

Therefor the follow-up of such infrastructure is particularly important to develop new guidelines or legislations. For this purpose, the 3 components of the water balance have been monitored on the BGW. This experiment has been particularly focused on a significant drained area collecting only green roof contribution (3511 m$^2$). The implemented set-up is described in the following.

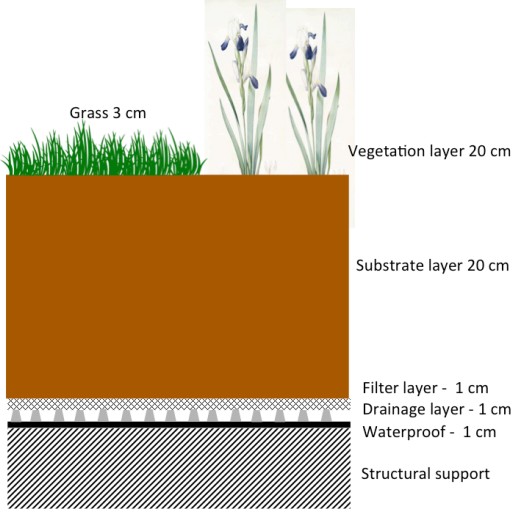

**Figure 2. Vertical profile of the green wave structure**

**2.2 Devices**

**2.2.1 Rainfall measurement**


Local rainfall has been analysed with the help of an optical disdrometer Campbell Scientific$^®$ PWS100. This device is made of two receivers and a transmitter generating four laser sheets. By analyzing the signals received from the light refracted by each drop passing through the 40 cm$^2$ sampling area, the drop size and velocity are estimated. A rain rate can then be derived. Disdrometers are now considered 175 as a reliable rainfall measurement instrument (Frasson et al., 2011; Gires et al., 2016; Thurai et al., 2011). The device has been installed since September 2013 on the roof of the Ecole des Ponts ParisTech building (see Fig.1). This disdrometer and its corresponding data have already been presented in details in a previous data paper (Gires et al., 2018) that summarizes a measurement campaign that took place in January-February 2016. Here, the rainfall data provided by this 180 disdrometer and characterized by a time step of 30 seconds has been used.

**2.2.2 Water content measurement**

Estimation of soil moisture represents a difficult challenge, as it deals with a highly spatially and temporally variable process (Lakshmi et al., 2003), essentially due to soil type and depth. Hence, 185 suitable systems are required to properly assess soil moisture. Nowadays a large number of sensors based on different methods are available for this purpose (Jackson et al., 2008). Among them, indirect methods based on electromagnetic (EM) principles have gained wide acceptance over the last decades. EM sensors have the advantage to deliver fast, in-situ, non-destructive and reliable measurements with acceptable precision (Stacheder et al., 2009).


Here Time Domain Reflectometry technique (TDR also known as capacitance) has been selected. It is an EM moisture measurement that determines an electrical property called electrical conductivity or dielectric constant ($k_a$). It is based on the interaction of an EM field with the soil water by using capacitance/frequency domain technology (Stacheder et al., 2009). The TDR sensor measures the 195 propagation time of an EM pulse, generated by a pulse generator and containing a broad range of different measurement frequencies. The electrical pulse is applied to the waveguides (traditionally a pair of parallel metallic rods) inserted in the soil. The incident EM travels across the length of the waveguides and then is reflected back when it reaches the end of the waveguides. The travel time required for the pulse to reach the end of the waveguides and come back depends on the dielectric 200 constant of the soil.

$$k_a = \left( \frac{c \cdot \Delta t}{2 \cdot L} \right)$$ (2)

Where $k_a$ is the bulk soil dielectric permittivity [-], $L$ the effective probe length [m] $\Delta t$ is the two-way travel time along the probe (s), and $c$ the velocity of EM wave in free space ($c=2.298\times10^8$ m/s)

It is then possible to estimate soil moisture content by analyzing the dielectric constant changes into the soil. The usual relationship between volumetric water content and dielectric constant is known as Topp's Equation (Topp et al., 1980). It is adapted to a homogeneous conventional soil. Note that this substrate can be considered as coarse enough to not clearly show the dielectric behaviour of a typical volcanic media (see (Palla et al., 2009b) for a similar assumption). For this reason, it is assumed the

dielectric constant–water content relationship does not exhibit a significant different from the Topp equation:

$$\theta = -5.3\times10^{-2} + 2.92\times10^{-2}k_a - 5.5\times10^{-4}k_a^2 + 4.3\times10^{-6}k_a^3$$ (3)

where $\theta$ is the volumetric soil water content [$m^3.m^{-3}$].


As an alternative to Topp equation, an additional study was conducted to assess this relationship in lab. Here, for information, the calibration curve obtained with compaction representing better the current condition is displayed. This compaction artificially was mimicked by applying vibrations (this causes the segregation of the material similar to what occurs in situ during a long period of time).


$$\theta = -3.01\times10^{-1} + 1.13\times10^{-1}k_a - 5.81\times10^{-3}k_a^2 + 9.85\times10^{-5}k_a^3$$ (4)

Given that the dielectric data is provided, potential users are free to use Topp's Equation as done in this paper, or another one.

Consequently, an ubiquitous wireless TDR sensor network has been implemented on the ENPC Blue Green Wave to measure both water content and temperature. For this purpose 32 CWS665 wireless TDR sensors (produced by Campbell Scientific®) were initially installed. The data were collected by 4 CWB100 wireless bases, able to store each the data of 8 sensors. Then the data was transferred to a data-logger CR6 from Campbell Scientific®. The initial selected time step was 1 minute. It appeared

that this first configuration was responsible for many gaps in the time series due to interferences between the different TDR sensors and the bases. To avoid this problem, only 16 TDR sensors were used, all of them connected to the same CWB100 base. For this same reason of possible interferences between the sensors, the time interval was enlarged to 4 minutes. Indeed, it is recommended to leave 15 seconds to ensure the connection of one sensor to the base. The final network aimed to capture the

space-time variability of water content in a heterogeneous soil as the BGW substrate. It was particularly adapted to assess the influence of the slope on infiltration and evapotranspiration processes.

**2.2.3 Discharge measurement**


Direct discharge measures are difficult to obtain in drainage pipes. For this reason, indirect measures using water level measurements are usually carried out. Here, water level inside the pipes was measured by a UM18 ultrasonic sensor (SICK, 2018) produced by SICK®. This sensor has been especially developed to perform non-contact distance measurement or detection of objects. The sensor

head emits an ultrasonic wave and receives the wave reflected back from the target. Ultrasonic sensors measure the distance to the target by measuring the time between the emission and reception. Implemented face to the water surface, it also measures the variation of the water level. The UM18 sensor is characterized by a nominal range of 250 mm, and an accuracy of 1% on this measurement range. For UM18 ultrasonic sensor, the dead zone is estimated to 5 mm. As the sensor has been placed

on the top of the conduit, only very high values (higher than 240 mm) could be affected by this dead zone. Since its implementation, water levels have never been higher than 120 mm.

One UM18 sensor has been implemented inside a pipe located in the garage in the building basement (see Figure 1). With a diameter of 300 mm, this pipe collects the water coming from a large part of the

BGW (approximately 1143 $m^2$). A standard 4–20 mA current loop is used to monitor or control

remotely these analogue sensors. The current is then transformed in voltage by a resistance of 100 Ω. The resulting transmitted signal ranges 400-2000 mV. In order to translate the electric signal in water level values, the following relationship has been applied

$$H_0 = (U - 460) \times \frac{250}{1600} \tag{5}$$

$H_0$ is the water level in mm, $U$ the measured voltage in mV, 460 represents the offset, 250 the modified nominal range in mm, 1600 the nominal range in mV.

The water level is then transformed into discharge by using the Manning-Strickler equation (Eq. 5). This formula is usually used to estimate the average velocity (and discharge) of water flowing in an open channel. It is commonly applied in sewer design containing circular pipes.

$$Q_0 = V \times S = K \times R^{\frac{2}{3}} \times i^{\frac{1}{2}} \times S \tag{6}$$

Where $V$ is the average water velocity [m.s$^{-1}$], $K$ the friction coefficient [-], $S$ the wet surface [m$^2$], $R$ the hydraulic radius [m], and $i$ the pipe slope [m/m] which is equal to 0.0074 here. $R$ and $S$ are directly linked to the water level:

$$R = \frac{S}{P} \tag{7}$$

$$S = \frac{(\theta - \sin(\theta)) \times r^2}{2} \tag{8}$$

$$P = r \times \theta \tag{9}$$

$$\theta = 2 \times \arccos\left(\frac{r - H}{r}\right) \tag{10}$$

$K$ has been chosen to 85. This value corresponds with a cast iron material.

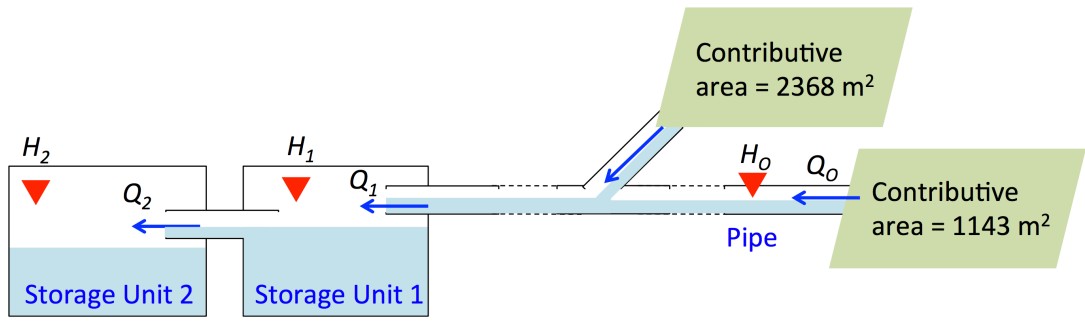

**Figure 3. Location of the water level sensors in the stormwater management network**

Two additional UM18 sensors have been implemented in the two consecutive storage units (see Figure 3) collecting the rainwater drained by a large contributive area of 3511 m$^2$, and including the previous monitored area. The first storage unit is a rainwater tank (characterized by a floor area of 32.2 m$^2$) devoted to irrigation. Filled most of the time, the excess water is routed by a pipe toward the second unit (floor area of 22.5 m$^2$). A relationship similar to Eq. 4 between the voltage measurement and the water level has been adjusted for both units:

$$H_i = (U - 0.38) \times \frac{20}{1.62} - dh \tag{11}$$

Here $U$ the measured voltage in V, the nominal range is 20 cm and $dh$ (equal to 1.06 cm) corresponds to an additional offset due to the elevation of the sensor

By studying both water level variations, a relationship between the water level measured in the first unit ($H_1$) and the outflow routing to the second unit $Q_2$ (and related to $H_2$) has been established (see

Figure 4). Finally, the total discharge reaching the first unit and collecting the downstream rainfall can be assessed by the following equation depending only on $H_1$:

$$Q_1 = Q_2 + \frac{dH_1}{dt} \times A1 = f(H_1) + \frac{dH_1}{dt} \times A1 \qquad (12)$$

where $Q_1$ is the discharge reaching the first unit and $Q_2$ the second respectively, $A1 = 33.2$ m$^2$ is floor area of the first unit.

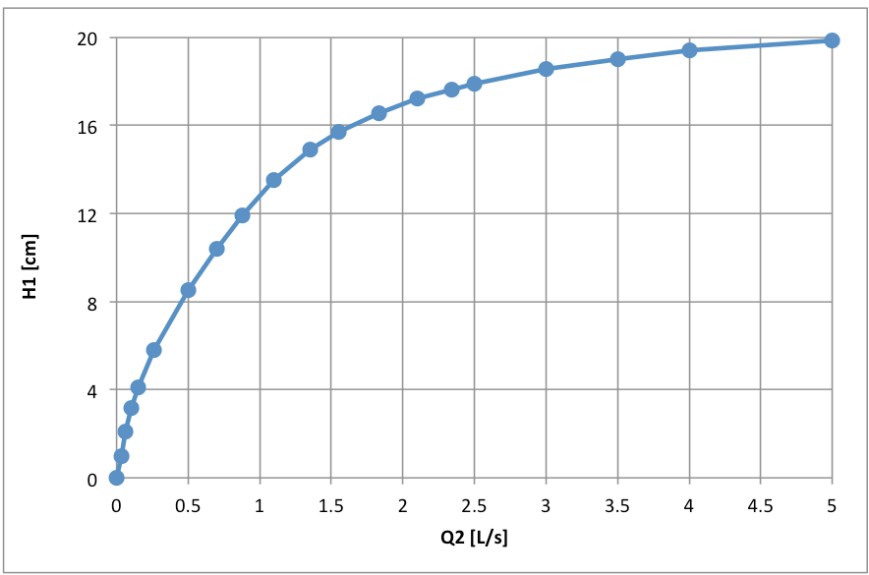

**Figure 4. Relationship adjusted between the water level $H_1$ and the downstream discharge $Q_2$**

Finally, discharge data was recorded with a time step of 30 seconds for the sensor implemented in the conduit, and 15 seconds for the one in the storage unit.

**2.3 Available output, data processing and period of study**

As already presented in detail in (Gires et al., 2018), precipitation data is collected in real time and stored through daily files.. Here, these files for 30 s time step rain rate have been gathered with the help of a Python script to create a long time series covering the whole period of study. Each line contains the time step expressed as YYYY-MM-DD HH:MM:SS and the corresponding rainfall intensity (in mm/h) separated by a comma.

Water content and water level data inside the pipe are collected and stored every night on the HM&Co server in two different files. For this purpose, the Loggernet software produced by Campbell Scientific® has been used. It supports programming, communication, and data retrieval between data loggers and a PC. Concerning the water level file, each line corresponds to a time step for which the following information is recorded (in each line, these values are separated by a comma):
- Exact definition of the time step expressed as YYYY-MM-DD HH:MM:SS
- Item number
- Voltage indicator to ensure the quality of the measurement (it should be close to 12 V)
- Internal temperature of the datalogger box
- Unused data coming from a non operational sensor
- Water level measured inside the pipe ($U$ in Eq. (4), expressed in mV)
- Unused data coming from a non operational sensor
- Unused data coming from a non operational sensor

Similar format has been chosen for volumetric water content data (note that names of the 16 VWC sensors are indicated in the header and also are reported on Figure 1):
- Exact definition of the time step expressed in YYYY-MM-DD HH:MM:SS
- Item number
- Voltage indicator to ensure the quality of the measurement (it should be close to 12 V)

335       - Internal temperature of the datalogger box
- Volumetric water content (expressed as $k_a$) for the 16 TDR sensors
- STT_B3: Summary Transfer Time for basis, which is related to the total time required for collecting information from all the sensors that are collected to that base.

Water level data inside the storage units have been collected by using the open-source Arduino Uno microcontroller board that works in the offline regime. This Arduino system was chosen because the storage unit was instrumented few months after the conduit, and that the distance was too long to make a connection between the storage unit and the existing data logger. Data are continuously stored on the 64 MB memory card implemented on the board, and copied manually to the HM&Co server once per
week. Data contain the following information (in each line, these values are separated by a space):
- Item number
- Voltage values for the first storage unit – *U1* (in mV)
- Voltage values for the second storage unit – *U2* (in mV)
- Exact definition of the time step expressed in YYYY-MM-DD HH:MM:SS
By using Equation (10) *U1* values are transformed into *H1* as a part of post-processing. Note that *U2* data have been used only for a short period of time after the implementation of UM18 sensors, until *Q2* = f(*H1*) functionality has been obtained. After that they were no longer necessary.

**3 Data availability**


Contrary to rainfall and discharge, which are measured continuously at the same locations, water content sensors can be moved from one location to another on the BGW. Moreover, they were rarely kept installed during the night for security reason. Nevertheless, during several months at the beginning of 2018, they were maintained on the same section of the BGW (the one showed in Figure 1). This time
period corresponds to 78 days, from February 19th to May 7th 2018. After this period, the water content sensors were moved to proceed to several evapotranspiration measurements campaigns on the BGW (see Conclusion section). The period has been selected to provide water balance components measurements to potential users. This data set is available for download from the following web page (Versini et al., 2019): https://doi.org/10.5281/zenodo.3687775

**3.1 Presentation of the available data set**

This data set presented in details in the next section contains the following files:
- A rainfall file: 2018_0219-0507_Data_rainfall.csv
- A water content file: 2018_0219-0507_VWC.csv
- A water level inside the pipe file: 2018_0219-0507_Data_discharge.csv
- A water level in the storage file: 2018_0219-0507_Data_Arduino.csv
- A python script to select the data, transform the raw data in physical measurements and carry out some initial analysis.

In details, the python script is structured as follow:
- Time period selection: this part could be changed to select a study time period by choosing an initial and final date.
- Data selection and transformation: the data corresponding to this time period is selected in the
380        different files. Electric signals measured by the water level sensors are converted in water level (by using Eq. 4 and 10), then in discharge by using Manning-Strickler equation (Eq. 5) for the pipe and Eq. 11 for the storage unit. In order to smooth the erratic 15s-signal produced by storage unit measurements, the computed discharge data are averaged on a moving window, whose number of time steps can be modified. Dielectric constants measured by the
385        16 TDR sensors are converted in water content by using Topp equation (Eq. 3).
- Representation of the computed data: Several figures are plotted to illustrate the variation of the hydrological components in time. The first one represents the corresponding hydrographs for both discharges computed inside the pipe and in the storage unit. The second one synthetizes the water content measured by the 16 TDR sensors. In each figure, the
390        precipitation is drawn on an invert y-axis.
- Computation of runoff coefficients: runoff coefficient is the ratio between the total amount of precipitation (computed by multiplying the rain depth by the corresponding contributive area) and the total volume of water flowing through the monitored pipe or the storage unit. This value ranging 0 to 100% illustrate the capacity of the green roof to retain rainwater.

### 3.2 Presentation of the time series

During the available time period including half of winter and half of spring, it rained a total amount of 123.1 mm (see Figure 5). The rainfall file has no missing value, and 6 rainfall events can be defined. They correspond to periods with cumulative rainfall depths greater than 5 mm (separated by a dry period of at least 6 hours) that caused discharge in both pipe and storage unit: 7th March (9 mm), 11[th] March (9.7 mm), 17[th] March (7.5 mm), 27[th] and 28[th] March (13.9 mm), 9[th] April (9.6 mm), 29 and 30[th] April (23.5 mm). These events are obviously not representative of the full range of precipitation events in the area. Nevertheless, it has to be mentioned that since the BGW was monitored (2017), intense rainfall has never caused any flooding on the surface, nor pipe filling (the higher water level measured was about 12 cm).

Concerning the 16 VWC sensors, 5.6% of the time steps are considered as missing data. This is essentially due to 2 particular sensors that were out of service from 16[th] March to the end of the study time period. The 16 sensors follow the same dynamic, responding to the several rainfall events (see Figure 6). Water content measurements decrease simultaneously during two long dry periods, at the end of February and from mid-April to the beginning of May. The sensors show a significant spatial variability in terms of absolute values. These differences illustrate the heterogeneousness of the substrate profiles in terms of hydrological behaviour. This is due to the granular composition of the substrate but also to the wavy-form of the BGW. Indeed, the lowest values tend to refer to the upstream sensors, whereas the highest values tend to refer to the downstream ones. Note that the grain size distribution time evolution is difficult to assess. Only the loss of some small particles has been noticed in the conduits.

Discharge data is almost complete. Only one measurement is missing in the pipe and 0.2% of total amount of time steps for the storage unit. These missing data correspond to the short periods during which the manual collection of the data was carried out. Note that in order to avoid the loss of relevant data, this collection was done during a dry period. Over this time period of 78 days, runoff coefficient computed for both pipe and storage unit are equal to 70.6% and 71.1% respectively. These close values demonstrate the suitability of the monitored set-up. The missing water corresponds to the water retained by the substrate and the vegetation. It should be returned to the atmosphere by evapotranspiration. As already mentioned, Topp equation (Eq. 3) used to convert dielectric constant in water content could not be adapted to the specific substrate implemented on the BGW. For this reason, the dielectric constant data are provided, leaving the reader free to use another relationship to convert this data in water content.

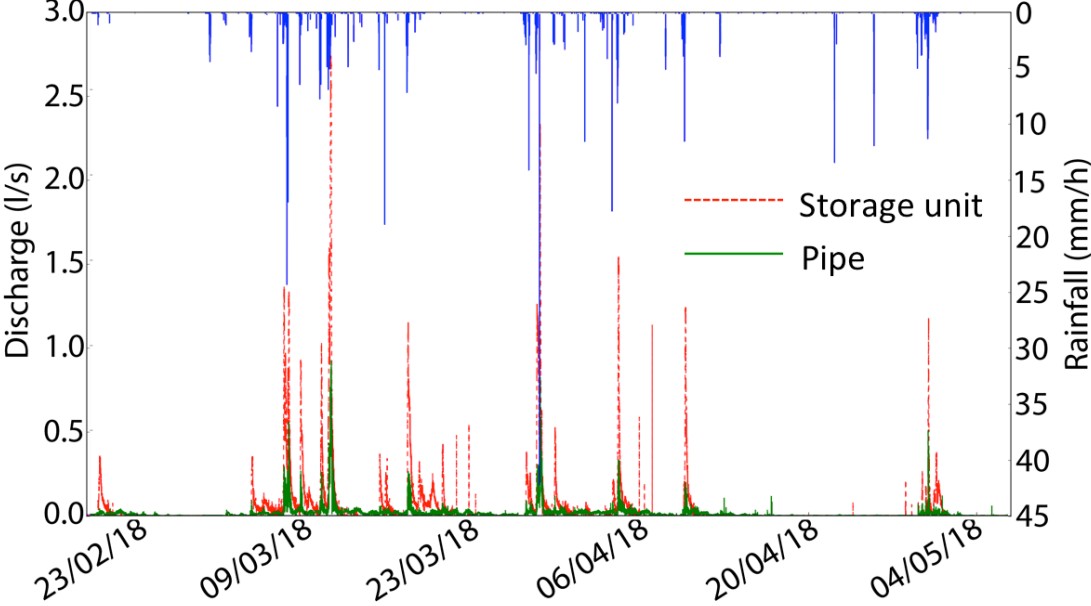

**Figure 5. Rainfall and computed discharges for the whole time period**

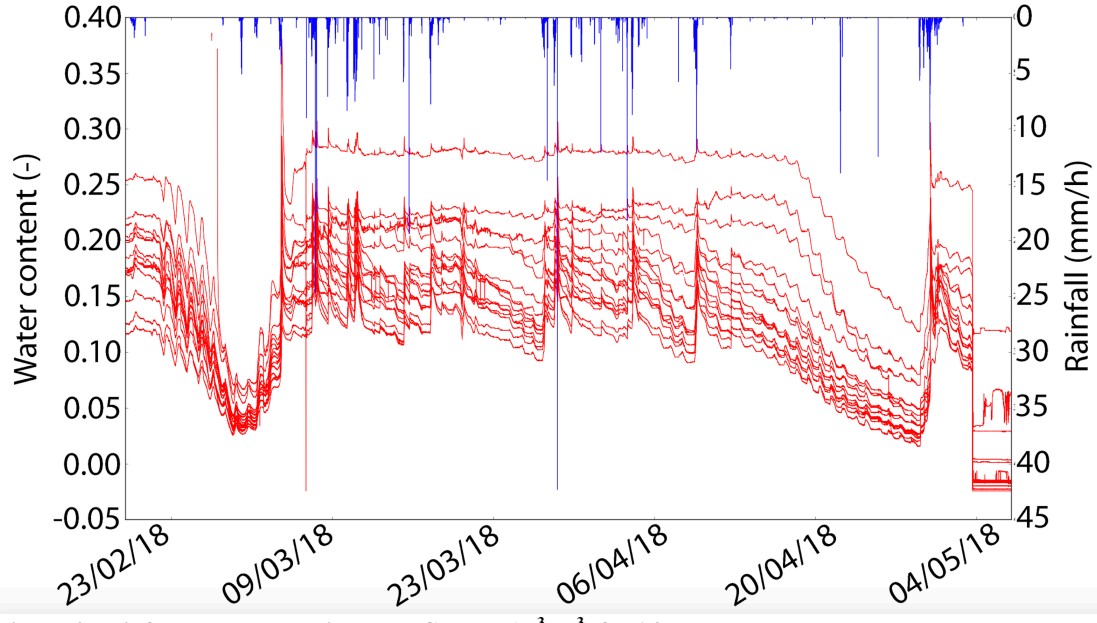

**Figure 6. Rainfall and Volumetric Water Content (m³.m⁻³) for 16 TDR sensors**

### 3.3 Illustration with a particular event

The 29 and 30th April rainfall event is presented in details in this section. It corresponds to the most intense event with a total cumulative rainfall depth of 23.5 mm. Figure 7 shows the corresponding hydrograph from which the delay between rainfall and discharge peaks can be deduced. It reaches 1h for the first contributive area (drained to the pipe) and 1.5 h for the second one (drained to the storage unit).

For a question of coherency with previous studies (Palla et al., 2009b for instance), the water content difference was computed with Topp's equation. The water stored in the substrate during this event was assessed by the difference between initial and final values. For the 16 sensors, this value ranges between 9.8% and 13.7%. This corresponds to a water depth comprised between 19.6 mm and 27.2 mm, and a storage capacity representing between 83% and higher than 100% of the rainfall (Note that a range comprised between 20.6 and 30.0 mm is obtained with the lab relationship presented in Eq. 4). It is clear the larger values are overestimated but the order of magnitude is consistent with the computed runoff coefficients: 15% for the surface drained to the pipe and 22% for the surface drained to the storage unit. This result illustrates the retention and detention properties of green roof. It has to be recalled that these impacts differ from one event to another depending on the precipitation and the initial conditions.

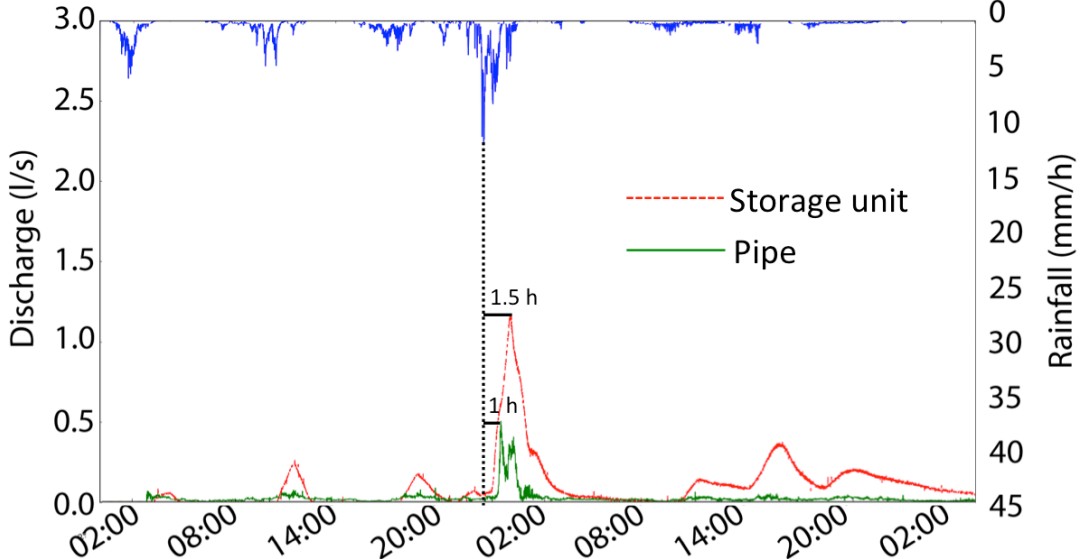

**Figure 7. Rainfall and computed discharges for the 29-30[th] April 2018 event**

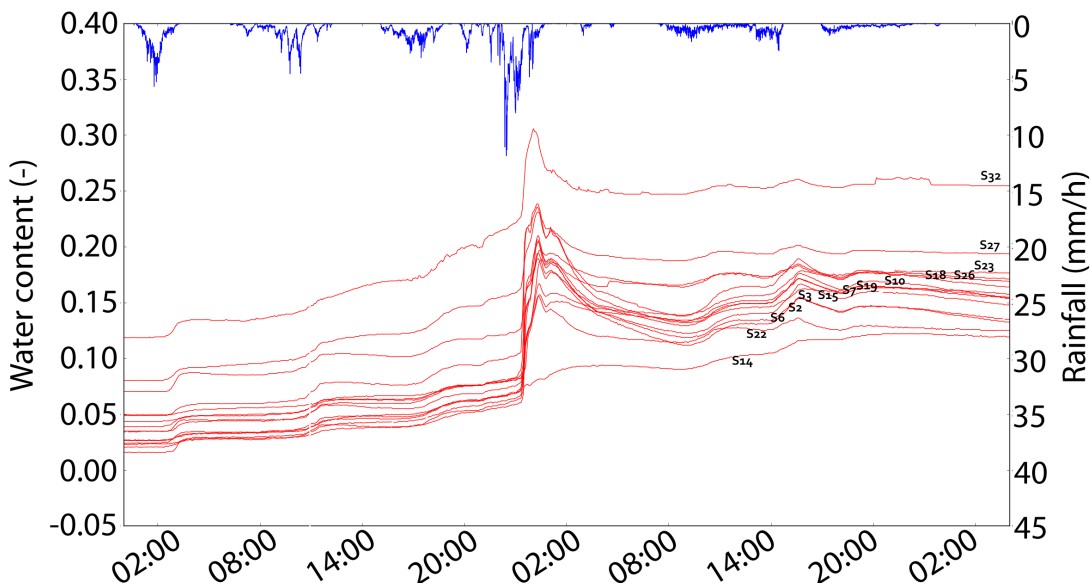

**Figure 8. Rainfall and Volumetric Water Content (m³.m⁻³) for 16 TDR sensors on the 29-30[th] April 2018 event (sensor references are indicating by increasing value at the end of the event)**

**4 Conclusion**

This paper presents the data collected by several devices devoted to the assessment of the water balance of a particular green roof located close to Paris. The dataset made available for research purposes contain 3 types of data, representing the relevant components of the water balance during a rainfall event: rainfall, water content in the substrate and the discharge flowing out of the infrastructure. These data were collected during 78 days between February and May 2018. These measurements are useful to study the capacity of such vegetated infrastructures to store rainwater and act as stormwater management tool. They could also be useful to develop and validate some appropriate modeling approaches (Stovin et al., 2013; Versini et al., 2016).

This data set is available for download free of charge from the following web page (Versini et al., 2019): https://doi.org/10.5281/zenodo.3687775

It is provided by the Hydrology, Meteorology, and Complexity laboratory of École des Ponts ParisTech (HM&Co-ENPC). The following references should be cited for every use of the data:

Versini, P.-A., Stanic, F., Gires, A., Scherzer, D., and Tchiguirinskaia, I. (2019). Measurement of the water balance components of a large green roof in Greater Paris Area. Earth System Science Data. XXXXX

Versini, P.-A., Stanic, F., Gires, A., Schertzer, D., Tchinguirinskaia, I.: Data for "Measurement of the water balance components of a large green roof in Greater Paris Area", https://doi.org/10.5281/zenodo.3467300, 2019

Researches focused on the assessment of ecosystem services provided by Blue Green Solutions is continuing at HM&Co-ENPC, and particularly on the BGW. The monitoring set-up has been recently extended to the energy balance components measurement (radiation balance, conduction, sensitive and latent heat flux) and particularly to the evapotranspiration flux. Such data will be particularly useful to study the ability of Blue Green Solutions to mitigate urban heat islands (but also to assess its retention potential during dry periods). The French ANR EVNATURB project (https://hmco.enpc.fr/portfolio-archive/evnaturb/), that aims to develop a platform to assess some of the eco-system services (ie stormwater management, cooling effect, or biodiversity conservation) provided by BGS is now pursuing this work of monitoring (Versini et al., 2017).

**Author contribution**
Pierre-Antoine Versini supervised the study, reviewed, and wrote a large part of the manuscript; Filip Stanic and Auguste Gires worked on the implementation of some of the presented sensors, the collection of the data and participate to the review of the paper; Daniel Schertzer and Ioulia Tchiguirinskaia collaborate to the study supervision and the review process.

**Competing interests**
The authors declare that they have no conflict of interest.

**Acknowledgment**
This work was initiated during the Climate-KIC funded Blue Green Dream project (http://bgd.org.uk/). It has also been supported by the Academic Chair "Hydrology for Resilient Cities", a partnership between Ecole des Ponts ParisTech and the Veolia group, and the ANR EVNATURB project dealing with the evaluation of ecosystem performance for re-naturing urban environment.

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
