# Peer review of "Measurements of the water balance components of a large green roof in Greater Paris Area"

_Earth System Science Data, 2019_

## Referee Comment (RC1) · Anonymous Referee #1 · 29 Nov 2019

General comments:

Whilst a large number of test bed scale monitoring studies of green roof hydrological performance exist in the literature already, there are relatively few examples of full scale monitoring. This data set is therefore a welcome addition.

If the intention is for other researchers or practitioners to make use of the data, e.g. for model validation, it is critically important that all details of the roof's configuration and monitoring are fully and clearly explained. As it stands, my confidence in the data quality is moderate – this can potentially be improved by some error estimation, and some analysis of individual storm-based data via rainfall-runoff hydrograph comparisons and mass balance volumetric comparisons. It is not clear whether the authors believe the moisture content data to be useable or not.

[Figure]

I also have some reservations about the completeness of the data. In particular, models of green roof retention require information about evapotranspiration. However it does not appear that climatic data required to estimate ET has been collected. This seriously compromises the data set's value for model validation.

Specific Comments:

1. Figure 1 is very difficult to understand, with photographs overlaid on top of a blurred engineering drawing. There are no dimensions. A clearer diagram is required.

2. Insufficient detail of the green roof profile is provided (line 136-7). Please provide a clear vertical section through the system, and confirm that it is consistent over the whole area. What is the actual shape/configuration of the drainage layer?

3. The information provided here on the physical properties of the substrate (lines 137-9) is insufficient for the validation of physically-based models. Please provide more detail on the particle size distribution, moisture retention characteristics and hydraulic conductivity.

4. The monitored area appears to be only a portion (1143 + 2368 m2) of the total roof area (1 ha, line 119). Please indicate the monitored portions clearly on a revised version of Figure 1. Some indication of the longest flow path lengths from catchment boundary to outlet would be useful for interpreting/modelling runoff detention.

5. Line 135 mentions two types of vegetation. Please show where each type of vegetation occurs on a revised version of Figure 1.

6. In line 209 it is suggested that the moisture content probes were specifically located to study the influence of slope. Therefore, please provide some information about the slope at their location.

7. Moisture content probes. Section 2.2.2 contains a lot of information about the general principles of soil moisture measurement, suggesting (Equation 3) that a standard calibration equation for natural soils was applied. There is no indication that these were

calibrated for the specific substrate used here. Other green roof studies have repeat-edly emphasised the need to undertake substrate-specific calibrations. The statement later on (lines 385 to 387) is unclear, but suggests that perhaps you don't trust this data. Do the authors recommend use of this data set or not?

8. What are the estimated uncertainties associated with discharge measurements (page 6)?

9. How did you define a storm event? Line 365. Is this based on a standard inter-event dry period of e.g. 6 hours, or something else?

10. Text on line 52 suggests that evapotranspiration can be neglected during storm events. This is a reasonable assumption for short events in cool or temperate cli-mates. However, it may not be correct for longer events and/or hotter climates. In all cases though, ET is a critical component of the overall water balance, as it is ET that generates the roof's retention potential (initial losses) during dry periods. Do you have climate data that would enable ET0 to be estimated (e.g. from Penman-Monteith FAO56 equation)?

11. Given the emphasis on acquiring the complete water balance, it would have been nice to see some evidence that the collected data is capable of demonstrating mass balance by comparing the total volumes of rainfall x catchment area, volumetric change in soil moisture and runoff for several specific storm events. It would also be good to see one or two illustrative hydrograph comparisons over shorter time-scales (< 24 hours). Do you see initial losses after long dry periods? Do you see lag and attenuation of the peak runoff or not?

12. I have attempted to retrieve and process some of the data. Data retrieval was straightforward. As I am not a python user, I chose to work with the raw rainfall and pipe discharge data files. The .dat files were read into Excel as csv files, and the data format appears to correspond to the description in the paper. The rainfall data is consistent with Figure 4. However, I have some concerns about the pipe discharge data. Missing

data is not clearly indicated in the file. Without smoothing, the data appears noisy, and it doesn't appear to return to zero between events. Pipe slope is not provided in the paper, so Q cannot be independently verified. The Manning-Strickler formula applies to steady uniform flow; its application here for the measurement of time-varying discharge needs more justification.

Technical corrections:

English is generally acceptable, though there are many minor errors and it would benefit from further editing.

There is a problem with the typesetting of superscripts in lines 40-41.

Line 365 – Reference to Figure 3 should be Figure 4.

Line 373 – Reference to Figure 4 should be Figure 5.

---

## Author Comment (AC1) · 20 Dec 2019

Dear Referee,

First of all, we would like to thank you for the interest you have shown to our manuscript and your detailed review. Here are our answers and some proposals to improve our paper regarding your comments and suggestions.

Regarding your specific comments (1), (4), (5) and (6), Figure 1 will be modified by: (i) separating photographs and the blurred engineering (the useless part will be removed), (ii) adding a scale, (iii) adding some indications about the monitored area and flow path lengths, (iv) indicating the location of areas with plants, (v) adding a profile of the section where the water content sensors were implemented (with the slopes).

[Figure]

Regarding your specific comments (2) and (3), some information will be added to better describe the green roof, including: (i) a figure presenting the vertical profile, (ii) more physical properties of the substrate. It is recalled that most of these properties and grain size distribution, water retention and hydraulic conductivity curves are available in Stanic et al., 2019.

Stanic, F., Cui, Y.J., Versini, P.-A., Schertzer, D., Tchiguirinskaia, I., 2019. A device for the simultaneous determination of the water retention properties and the hydraulic conductivity of an unsaturated green-roof material. Geotech. Test. J., 43(3), https://doi.org/10.1520/GTJ20170443 (https://compass.astm.org/DIGITAL_LIBRARY/JOURNALS/GEOTECH/PAGES/GTJ20170443.htm)

Specific comments (7): Indeed Topp equation (Eq. 3) has been defined for natural soil. Moreover it gives the link between the dielectric constant and the volumetric water content which is strongly related to the bulk density (compaction of the substrate). Here it can be assumed that the BGW substrate was coarse enough to not clearly show the dielectric behaviour of a typical volcanic media, and do not reveal a dielectric constant–volumetric water content relation significantly different from the Topp equation (see Palla et al., 2009 for a similar assumption). Nevertheless such assumption can conduct to the under-estimation of water content. Additional study was done to assess this relationship in lab. Two different calibration curves were obtained: (i) without applying any compaction force (lower bulk density, higher porosity), (ii) with compaction obtained by applying vibrations (this causes the segregation of the material similar to what occurs in situ during a long period of time). When comparing with the Topp equation, these curves show that the transformation from dielectric constant into the volumetric water content is not straightforward. For this reason, we provide the dielectric constant data, letting free the reader to use another relationship to convert this data in water content. Additional relationships can be added in the manuscript.

The statement "The sensors show a significant spatial variability in terms of absolute values. These differences illustrate the heterogeneousness of the substrate, due to its

granular composition and its wavy-form" means the sensors were accurate enough to measure different water content behaviours generating by some different hydrological behaviours (due to the slope and different vertical profiles). By using Topp equation or another relationship, the sensors provide some relevant information about this spatial variability.

Specific comments (8): Concerning the water level sensor, the uncertainly represents +/- 1% of the measured value. The resulting uncertainty for discharge estimate could be assessed from general formula of uncertainty propagation.

Specific comments (9): Here, the rainfall events have been defined by analysing both rainfall and discharge data. Following the standard inter-event dry period of 6 hours, 6 rainfall events characterized by a total amount higher than 5 mm can be defined: 7th March (9 mm), 11th March (9.7 mm), 17th March (7.5 mm), 27th and 28th March (13.9 mm), 9th April (9.6 mm), 29 and 30th April (23.5 mm). This can be modified in the manuscript.

Specific comments (10): In Paris region (temperate climate), evapotranspiration can be neglected in the water balance during a rainfall event. Hot temperature can occur in summer, but storms are very short in such situation. We completely agree that evapotranspiration represents a key information to assess the retention capacity of the substrate during dry periods. In reality, the monitoring set-up of the BGW has been recently extended to the energy balance components measurement and particularly to the evapotranspiration flux. It is explained in the last paragraph of the manuscript dedicated to perspectives. For now, the evolution of the retention capacity of the substrate during dry periods is assessed by the water content sensors.

Specific comments (11): It is possible to add a particular rainfall event to present the mentioned discharge (the reader can do it with the python script). Indeed, green roof impacts (runoff coefficient, lag and attenuation of the peak runoff) differ from one event to another depending on the precipitation but also the initial conditions. Specific comments (12): Concerning the discharge measures inside the pipe, only one data is missing (2018-04-28 14:58:33). The slope is equal to 0.74% (it was indicated in the Python script, it should also be the case in the manuscript). It is true that Manning-Strickler equation is valid under the assumption of uniform flow. Here the very low slope (inducing low water velocities) and the absence of connection before and after the location of the sensor makes relevant the use of the Manning-Strickler formula (the flow profile is close to uniform). This assumption is usually done in stromwater management (see SMWW model developed by the Environmental Protection Agency in USA for instance).

Concerning the discharge measures from the storage units, there are 74 time steps characterized by NaN values in the file (19/02/18 at 17:06 and the following for instance). As mentioned in Section 3.1 the 15s-signal produced by the sensor is very erratic. In order to smooth this signal, the data can be averaged on a moving window (whose number of time steps can be modified in the Python script).

The Authors

---

## Referee Comment (RC2) · Anonymous Referee #2 · 4 Feb 2020

Review ESSD-2019-187

Very interesting data set, unusual perhaps for ESSD but of potential high value none-the-less.

Data download easily from Zenodo. Good data organisation and formats, easy to open and use both the .dat file and the Python scripts. Good metadata headers in discharge and VWC files but missing (and much needed - you do not want users like me guessing at the data columns) in Arduino and rainfall files. In downloading MacOSX versions I found that changing suffixes from .dat to .csv made files much easier to use in text editors, spreadsheets, GIS software, etc. Consider .csv rather than .dat? Or include a hint for users about changing .dat to .csv?

Overall, with small improvements in metadata, a good useful data product. Presentation however remains weak. Improvements in descriptions and narrative will help many users.

Overall comments:

- The manuscript highlights large areal coverage (e.g. 1 ha) of BGW but in fact the data only cover 3k $m^2$ (e.g. figure 2 and line 252). The area under measurement here still exceeds prior studies by at least a factor of 5, but advertised 1 ha (10k $m^2$) while having data for 'only' 3k $m^2$ seems misleading or perhaps even dishonest?

- The time period of this data set (2018 February to May) misses the usual period of heaviest rainfall for Paris: intense afternoon late-afternoon thunderstorms in mid- to late-summer? Impressive that these authors achieved such high data collection rates (e.g. section 3.2 on times series performance) but do they contend that these measurements cover the full range of precipitation events? If not, they should inform readers about context of these particular months. What would happen (has happened) in heavy (rain rates greater than 20 mm / h) summer rainfall events? Soil / substrate erosion? Aerial flooding? Storage unit 1 fills and overflows to storage unit 2? Ultrasonic proximity/distance sensors in pipe or in storage unit 1 get immersed? Why did the measurements end in May 2018? Particularly curious about this statement at line 381: "this operation is done during a dry period".

- The authors rightly give high attention to retention / detention issues: water storage and run-off delays due to BGW. But, unfortunately, nowhere does a user find hints that these data might actually allow one to calculate retention or detention. Data providers know area, substrate, depth to impervious layer, soil moisture content, rainfall inputs, etc. But they leave it to users to try to calculate e.g. retention? Or they leave the impression that, despite qualify of measurements, one can not actually derive retention / detention? E.g at the time resolution used in figure 4, discharge looks simultaneous / instantaneous with rainfall. The system provides no detention? Or, the data do not allow user to calculate detention. Having raised the issue often and prominently in the introduction and justification, the authors seem remiss to not address whether their data prove relevant to those questions? Give us an example or address what one would need differently or additionally to actually calculate the BGW impacts on retention / detention? We see reference to these values (as outputs from the Python scripts) at lines 357 to 360 but the authors should give us a graphic example with specified uncertainties? Does the system actually produce useful numbers?

Specific comments:

The manuscript needs many small changes / improvements in language. Proofreading will catch many but not all of these errors. I record several specific concerns below. No doubt I missed others.

Line 43 "reaching the network". I believe the authors refer here to the stormwater management network but - unfortunately - the manuscript displays too many possible terms and explanations: sometimes 'network', sometimes 'rainfall network', sometimes 'stormwater network', rarely 'stormwater management network'. Settle on and define a standard language, then use it throughout.

Line 140, figure 1: In the upper right the figure lists 32 soil moisture sensors but - at left center of the figure and in text lines 200 to 210 - the authors show and explain use of only 16 sensors. Make 16 sensor the default configuration with parenthetical note or footnote about why 32 sensors seemed to exceed logger bandwidth? Fix sensor number in figure 1?

Line 221: "a nominal range of 250 mm". Clever to use ultrasonic distance / proximity sensors to measure water height but most ultrasonic sensors have dead zone or null zone close to the sensor face. Data sheets for ultrasonic sensors often specify "little or no dead zone" but more careful analysis suggests working dead zone of 2 cm. This represents nearly 10% of the working range of the UM18. Can manufacturers or authors certify linear response outside of that dead zone out to the maximum range? Have authors in this case relied entirely on manufacturer data sheets? If so, tell the user? Do the ultrasonic sensors, particularly in the pipe or in storage unit 1, get wet or get immersed? What happens then? Why do some file names include the term 'Arduino' (which I know well)? Arduino MPU to control the UM18, sending serial data to Campbell data logger? Or, does Arduino refer to the "Unused data coming from a non operational sensor." Evidently the term 'Arduino' applies to storage data but not pipe data? Sensor operated differently or data recorded differently in the two situations?

Line 247 and 252, figure 2: total contributive area of 3511 m$^2$. See comment above about measured area vs total area.

Line 272 figure 3: If x axis legend of this figure is correct (e.g. Q2 in liter / second) as I think it is, then figure legend ("downstream discharge Q1") seems wrong? Should read 'downstream discharge Q2'?

Line 375, 376: "heterogeneousness of the substrate, due to its granular composition and its wavy-form". Perhaps, but also including sensor-level uncertainties / imprecisions in measuring soil moisture? To the extent "granular composition" and "wavy-form" have an influence, do those represent features of the original BG roof or features that have evolved during time of existence?

---

## Author Comment (AC2) · 11 Feb 2020

Dear Referee,

First of all, we would like to thank you for the interest you have shown to our manuscript and your detailed review. Here are our answers and some proposals to improve our paper regarding your comments and suggestions.

Comments concerning the files: Some headers can easily be added in both Arduino and rainfall files to recall data source and units. The format conversion in .csv can also be considered.

Overall comment (1): The present water level sensors measure the discharge flowing out of a third of the BGW (3,500 m2 on the 10,000 m2). It is already significantly

larger than any prior studies (see references in Introduction section). Moreover, water content sensors can be moved (and have already been moved) over the BGW for additional purposes, as it was the case for evapotranspiration measurements (not presented here). In fact, the whole BGW is used as a pilot site for Blue green solution assessment. Some clarifications should be added in BGW presentation to avoid any misunderstanding and clarify the content of what is presented here as well as the overall context.

Overall comment (2): Indeed, in May 2018 the water content sensors were moved over the BGW to proceed to several evapotranspiration campaigns (comparison with the measurements made with an evapotranspiration chamber on a small area). For this reason, the continuous dataset ends in May 2018. The authors are aware the 6 presented rainfall events are not representative of the full range of precipitation events. Nevertheless, it has to be mentioned that since the BGW is monitored (2017), intense rainfall has never caused any flooding on the surface, nor pipe filling (the higher water level measured was about 12 cm).

The statement at line 381: "this operation is done during a dry period" refers to the collection of Arduino data. Arduino data are currently collected manually. During this operation the sensor is disconnected and no measurement is recorded. To avoid the possible loss of relevant measurements, this collection procedure is carried out during "dry periods" characterized by no rainfall and discharge.

All this information will be added to better understand the context in which this dataset was made.

Overall comment (3): BGW detention and retention properties differ from one event to another depending on the precipitation but also the initial conditions. Detention can be graphically seen at the rainfall event scale (usually 45 minutes between peak rainfall and peak discharge) and retention by computing the runoff coefficient. Both can be done by using the proposed Python script. As mentioned in response to reviewer 1,

the user has to be cautious concerning water retention estimation. We propose to use Topp equation to convert dielectric constant in water content, but we are aware of the possible weakness of this assumption (see response to reviewer 1). Finally, we provide the dielectric constant data, letting free the reader to use another relationship to convert this data in water content. Nevertheless, it is possible to add a particular rainfall event in the paper to illustrate the possible hydrological impacts of the BGW, and compute the runoff coefficient for every water content measurement for an example to illustrate the opportunities offered by this data set. Specific comments (1): "stormwater management network" should be the appropriate terminology. The text will be homogenised to avoid any confusion.

Specific comments (2): As commented in response to reviewer 1, Figure 1 should be modified. Only 16 sensors will be presented.

Specific comments (3): For UM18 this dead zone is estimated to 5 mm in the datasheet. As the sensor is placed on the top of the conduit, only very high values (higher than 240 mm) could be affected by this dead zone. For this range [0-240 mm], the measures made by the sensor were manually verified with some standards. Note that water levels have never been higher than 120 mm for now. For both conduit and storage units, the ultrasonic sensors have never been immersed.

Indeed, Arduino refers to the data collected in the storage unit. Campbell data logger collects only the data measured inside the pipe. The reason for which there are two different record systems is due to the fact that the storage unit was instrumented few months after the conduit, and that the distance was too long to make a connection between the storage unit and the existing data logger.

All these precisions will be added in the manuscript.

Specific comments (4): Some additional information will be added to avoid any confusion.
Specific comments (5): Indeed, it should be indicated "downstream discharge Q1 " in the figure legend.

Specific comments (6): For sure, sensor-level uncertainties can be added as an explanation of this spatial variability. The wavy-form has not moved during time as it was an architectural choice, and the roof is included in the concrete structure of the building. Concerning the granular composition, the natural grain size distribution of the substrate (see Stanic et al., 2019) can explain for a large part this spatial variability. It is quite difficult to assess how it has evolved with time. We have only notices that some of the small particles have been drained out of the substrate.

The authors

---

## Author Response (AR1)

Dear Editor,

First of all, we would like to thank both reviewers for their interest to our manuscript and their detailed reviews. Some efforts have been made to improve our paper regarding these comments and suggestions.

**Reviewer 1**

1. Figure 1 is very difficult to understand, with photographs overlaid on top of a blurred engineering drawing. There are no dimensions. A clearer diagram is required.
4. The monitored area appears to be only a portion (1143 + 2368 m2) of the total roof area (1 ha, line 119). Please indicate the monitored portions clearly on a revised version of Figure 1. Some indication of the longest flow path lengths from catchment boundary to outlet would be useful for interpreting/modelling runoff detention.
5. Line 135 mentions two types of vegetation. Please show where each type of vege- tation occurs on a revised version of Figure 1.
6. In line 209 it is suggested that the moisture content probes were specifically located to study the influence of slope. Therefore, please provide some information about the slope at their location.

Regarding specific comments (1), (4), (5) and (6), Figure 1 has been modified: (i) photographs and the blurred engineering drawing have been separated and simplified (on which a scale has been added), (ii) the monitored area and some indications about flow path lengths have been added, (iii) areas with plants have been indicated, (iv) a profile of the section where the water content sensors were implemented (indicating the slopes) has been added.

2. Insufficient detail of the green roof profile is provided (line 136-7). Please provide a clear vertical section through the system, and confirm that it is consistent over the whole area. What is the actual shape/configuration of the drainage layer?
3. The information provided here on the physical properties of the substrate (lines 137- 9) is insufficient for the validation of physically-based models. Please provide more detail on the particle size distribution, moisture retention characteristics and hydraulic conductivity.

Regarding specific comments (2) and (3), some information has been added to better describe the green roof: (i) a figure (new Figure 2) presenting the vertical profile has been added, (ii) more physical properties of the substrate (Table 1 + previous paragraph) have been provided (It has been recalled that grain size distribution, water retention and hydraulic conductivity curves are available in Stanic et al., 2019).

7. Moisture content probes. Section 2.2.2 contains a lot of information about the general principles of soil moisture measurement, suggesting (Equation 3) that a standard calibration equation for natural soils was applied. There is no indication that these were calibrated for the specific substrate used here. Other

green roof studies have repeatedly emphasised the need to undertake substrate-specific calibrations. The statement later on (lines 385 to 387) is unclear, but suggests that perhaps you don't trust this data. Do the authors recommend use of this data set or not?

Specific comment (7): Indeed Topp equation (Eq. 3) has been defined for natural soil. Moreover it gives the link between the dielectric constant and the volumetric water content which is strongly related to the bulk density (compaction of the substrate). Here it can be assumed that the BGW substrate was coarse enough to not clearly show the dielectric behaviour of a typical volcanic media, and do not reveal a dielectric constant–volumetric water content relation significantly different from the Topp equation (see Palla et al., 2009 for a similar assumption). This justification has been added in Section 2.2.2.

Nevertheless such assumption can also conduct to the mis-estimation of water content. Additional study was done to assess this relationship in lab. A calibration curve obtained with compaction made by applying vibrations has been added (Eq. 4). When comparing with the Topp equation, these curves show that the transformation from dielectric constant into the volumetric water content is not straightforward. For this reason, we provide the dielectric constant data, letting free the reader to use another relationship to convert this data in water content.

The statement "The sensors show a significant spatial variability in terms of absolute values. These differences illustrate the heterogeneousness of the substrate, due to its granular composition and its wavy-form" means the sensors were accurate enough to measure different water content behaviours generating by some different hydrological behaviours (due to the slope and different vertical profiles). By using Topp equation or another relationship, the sensors provide some relevant information about this spatial variability. This is explained in Section 3.2.

8. What are the estimated uncertainties associated with discharge measurements (page 6)?

Specific comment (8): Concerning the water level sensor, the uncertainly represents +/- 1% of the measured value. This information is written in Section 2.2.3 (first paragraph).

9. How did you define a storm event? Line 365. Is this based on a standard inter-event dry period of e.g. 6 hours, or something else?

Specific comment (9): Here, the rainfall events have been defined by analysing both rainfall and discharge data. Following the standard inter-event dry period of 6 hours, 6 rainfall events characterized by a total amount higher than 5 mm can be defined: 7th March (9 mm), 11th March (9.7 mm), 17th March (7.5 mm), 27th and 28th March (13.9 mm), 9th April (9.6 mm), 29 and 30th April (23.5 mm). This has been modified in the manuscript (Section 3.2).

10. Text on line 52 suggests that evapotranspiration can be neglected during storm events. This is a reasonable assumption for short events in cool or temperate cli- mates. However, it may not be correct for longer events and/or hotter climates. In all cases though, ET is a critical component of the overall water balance, as it is ET that generates the roof's retention potential (initial losses) during dry periods. Do you have climate data that would enable ET0 to be estimated (e.g. from Penman-Monteith FAO56 equation)?

Specific comment (10): In Paris region (temperate climate), evapotranspiration can be neglected in the water balance during a rainfall event. It has been specified in Introduction to avoid any confusion. Hot temperature can occur in summer, but storms are very short in such situation. We completely agree that evapotranspiration represents a key-information to assess the retention capacity of the substrate during dry periods. In reality, the monitoring set-up of the BGW has been recently extended to the energy balance components measurement and particularly to the evapotranspiration flux. It is explained in the last paragraph of the manuscript dedicated to perspectives. For now, the evolution of the retention capacity of the substrate during dry periods is assessed by the water content sensors.

11. Given the emphasis on acquiring the complete water balance, it would have been nice to see some evidence that the collected data is capable of demonstrating mass balance by comparing the total volumes of rainfall x catchment area, volumetric change in soil moisture and runoff for several specific storm events. It would also be good to see one or two illustrative hydrograph comparisons over shorter time-scales (< 24 hours). Do you see initial losses after long dry periods? Do you see lag and attenuation of the peak runoff or not?

Specific comment (11): A particular rainfall event is now presented in the new Section 3.3 (figures 7 and 8). It illustrates the ability of green roof to retain and detain rainfall water. The water balance is also computed by assessing the different terms. The water retained by the substrate (estimated by water content measurements) appears to be consistent with the runoff coefficient.

12. I have attempted to retrieve and process some of the data. Data retrieval was straightforward. As I am not a python user, I chose to work with the raw rainfall and pipe discharge data files. The .dat files were read into Excel as csv files, and the data format appears to correspond to the description in the paper. The rainfall data is consistent with Figure 4. However, I have some concerns about the pipe discharge data. Missing data is not clearly indicated in the file. Without smoothing, the data appears noisy, and it doesn't appear to return to zero between events. Pipe slope is not provided in the paper, so Q cannot be independently verified. The Manning-Strickler formula applies to steady uniform flow; its application here for the measurement of time-varying discharge needs more justification.

Specific comments (12): Concerning the discharge measures inside the pipe, only one data is missing (2018-04-28 14:58:33). The slope is equal to 0.74% (it was

indicated in the Python script, it is now also the case in the manuscript in Section 2.2.3). This very low slope and the absence of connection before and after the location of the sensor makes relevant the use of the Manning-Strickler formula. This assumption is usually done in stromwater management (see SMWW model developed by the Environmental Protection Agency in USA for instance). It is explained in the manuscript in Section 2.2.3.

Concerning the discharge measures from the storage units, there are 74 time steps characterized by NaN values in the file (19/02/18 at 17:06 and the following for instance). As mentioned in Section 3.1 the 15s-signal produced by the sensor is very erratic. On order to smooth this signal, the data can be averaged on a moving window (whose number of time steps can be modified in the Python script).

**Reviewer 2**

Data download easily from Zenodo. Good data organisation and formats, easy to open and use both the .dat file and the Python scripts. Good metadata headers in discharge and VWC files but missing (and much needed - you do not want users like me guessing at the data columns) in Arduino and rainfall files. In downloading MacOSX versions I found that changing suffixes from .dat to .csv made files much easier to use in text editors, spreadsheets, GIS software, etc. Consider .csv rather than .dat? Or include a hint for users about changing .dat to .csv?

Headers have been added in both Arduino and rainfall files to recall data source and units. .dat files have been replaced .csv files. Python program has been adapted to this new format.

Overall comments:

The manuscript highlights large areal coverage (e.g. 1 ha) of BGW but in fact the data only cover 3k m2 (e.g. figure 2 and line 252). The area under measurement here still exceeds prior studies by at least a factor of 5, but advertised 1 ha (10k m2) while having data for 'only' 3k m2 seems misleading or perhaps even dishonest?

The present water level sensors measure the discharge flowing out of a third of the BGW (3,500 $m^2$ on the 10,000 $m^2$). It is already significantly larger than any prior studies (see references in Introduction section). Moreover, water content sensors can be moved (and have already been moved) over the BGW for additional purposes, as it was the case for evapotranspiration measurements (not presented here). In fact, the whole BGW is used as a pilot site for Blue green solution assessment. Some clarifications have been added in BGW presentation (Section 2.1 and Conclusion) to avoid any misunderstanding and clarify the content of what is presented here as well as the overall context.

The time period of this data set (2018 February to May) misses the usual period of heaviest rainfall for Paris: intense afternoon late-afternoon thunderstorms in

mid- to late-summer? Impressive that these authors achieved such high data collection rates (e.g. section 3.2 on times series performance) but do they contend that these measurements cover the full range of precipitation events? If not, they should inform readers about context of these particular months. What would happen (has happened) in heavy (rain rates greater than 20 mm / h) summer rainfall events? Soil / substrate erosion? Aerial flooding? Storage unit 1 fills and overflows to storage unit 2? Ultrasonic proximity/distance sensors in pipe or in storage unit 1 get immersed? Why did the measurements end in May 2018? Particularly curious about this statement at line 381: "this operation is done during a dry period".

Indeed, in May 2018 the water content sensors were moved over the BGW to proceed to several evapotranspiration campaigns (comparison with the measurements made with an evapotranspiration chamber on a small area). For this reason, the continuous dataset ends in May 2018 (added in the first paragraph of Section 3). The authors are aware the 6 presented rainfall events are not representative of the full range of precipitation events. Nevertheless, it has to be mentioned that since the BGW is monitored (2017), intense rainfall has never caused any flooding on the surface, nor pipe filling (the higher water level measured was about 12 cm). This remark has been added in Section 3.2.

The statement at line 381 (now 418): "this operation is done during a dry period" refers to the collection of Arduino data. Arduino data are currently collected manually. During this operation the sensor is disconnected and no measurement is recorded. To avoid the possible loss of relevant measurements, this collection procedure is carried out during "dry periods" characterized by no rainfall and discharge. It has been mentionned.

The authors rightly give high attention to retention / detention issues: water storage and run- off delays due to BGW. But, unfortunately, nowhere does a user find hints that these data might actually allow one to calculate retention or detention. Data providers know area, substrate, depth to impervious layer, soil moisture content, rainfall inputs, etc. But they leave it to users to try to calculate e.g. retention? Or they leave the impression that, despite qualify of measurements, one can not actually derive retention / detention? E.g at the time resolution used in figure 4, discharge looks simultaneous / instantaneous with rainfall. The system provides no detention? Or, the data do not allow user to calculate detention. Having raised the issue often and prominently in the introduction and justification, the authors seem remiss to not address whether their data prove relevant to those questions? Give us an example or address what one would need differently or additionally to actually calculate the BGW impacts on retention / detention? We see reference to these values (as outputs from the Python scripts) at lines 357 to 360 but the authors should give us a graphic example with specified uncertainties? Does the system actually produce useful numbers?

BGW detention and retention properties differ from one event to another depending on the precipitation but also the initial conditions. Detention can be graphically seen at the rainfall event scale (usually 45 minutes between peak

rainfall and peak discharge) and retention by computing the runoff coefficient. Both can be done by using the proposed Python script.

As mentioned in response to reviewer 1, the user has to be cautious concerning water retention estimation. We propose to use Topp equation to convert dielectric constant in water content, but we are aware of the possible weakness of this assumption (see response to reviewer 1). Following both reviewers' comments, we have added a second relationship calibrated in lab. Finally, we provide the dielectric constant data, letting free the reader to use another relationship to convert this data in water content.

As also suggested reviewer 1, we have also addes a particular rainfall event in the paper to illustrate the possible hydrological impacts of the BGW, and compute the runoff coefficient for every water content measurement for an example to illustrate the opportunities offered by this data set.

Specific comments:

Line 43 "reaching the network". I believe the authors refer here to the stormwater management network but - unfortunately - the manuscript displays too many possible terms and explanations: sometimes 'network', sometimes 'rainfall network', sometimes 'stormwater network', rarely 'stormwater management network'. Settle on and define a standard language, then use it throughout.

"stormwater management network" should be the appropriate terminology. The text will be homogenised to avoid any confusion.

Line 140, figure 1: In the upper right the figure lists 32 soil moisture sensors but - at left center of the figure and in text lines 200 to 210 - the authors show and explain use of only 16 sensors. Make 16 sensor the default configuration with parenthetical note or footnote about why 32 sensors seemed to exceed logger bandwidth? Fix sensor number in figure 1?

As commented in response to reviewer 1, Figure 1 has been modified. Only 16 sensors are presented.

Line 221: "a nominal range of 250 mm". Clever to use ultrasonic distance / proximity sensors to measure water height but most ultrasonic sensors have dead zone or null zone close to the sensor face. Data sheets for ultrasonic sensors often specify "little or no dead zone" but more careful analysis suggests working dead zone of 2 cm. This represents nearly 10% of the working range of the UM18. Can manufacturers or authors certify linear response outside of that dead zone out to the maximum range? Have authors in this case relied entirely on manufacturer data sheets? If so, tell the user? Do the ultrasonic sensors, particularly in the pipe or in storage unit 1, get wet or get immersed? What happens then? Why do some file names include the term 'Arduino' (which I know well)? Arduino MPU to control the UM18, sending serial data to Campbell data logger? Or, does Arduino refer to the "Unused data coming from a non

operational sensor." Evidently the term 'Arduino' applies to storage data but not pipe data? Sensor operated differently or data recorded differently in the two situations?

For UM18 this dead zone is estimated to 5 mm in the datasheet. As the sensor is placed on the top of the conduit, only very high values (higher than 240 mm) could be affected by this dead zone. For this range [0-240 mm], the measures made by the sensor were manually verified with some standards. Note that water levels have never been higher than 120 mm for now. For both conduit and storage units, the ultrasonic sensors have never been immersed. This comment has been added in Section 2.2.3.

Indeed, Arduino refers to the data collected in the storage unit. Campbell data logger collects only the data measured inside the pipe. The reason for which there are two different record systems is due to the fact that the storage unit was instrumented few months after the conduit, and that the distance was too long to make a connection between the storage unit and the existing data logger. This has been specified in Section 2.3.

Line 247 and 252, figure 2: total contributive area of 3511 m². See comment above about measured area vs total area.

Additional information has been added in Section 2.1 and in Figure 1 to avoid any confusion.

Line 272 figure 3: If x axis legend of this figure is correct (e.g. Q2 in liter / second) as I think it is, then figure legend ("downstream discharge Q1") seems wrong? Should read 'downstream discharge Q2'?

In fact, it should be indicated "downstream discharge Q2 " in the figure caption.

Line 375, 376: "heterogeneousness of the substrate, due to its granular composition and its wavy-form". Perhaps, but also including sensor-level uncertainties / imprecisions in measuring soil moisture? To the extent "granular composition" and "wavy-form" have an influence, do those represent features of the original BG roof or features that have evolved during time of existence?

For sure, sensor-level uncertainties can be added as an explanation of this spatial variability. The wavy-form has not moved during time as it was an architectural choice, and the roof is included in the concrete structure of the building. Concerning the granular composition, the natural grain size distribution of the substrate (see Stanic et al., 2019) can explain for a large part this spatial variability. It is quite difficult to assess how it has evolved with time. We have only noticed that some of the small particles have been drained out of the substrate. It has been specified in Section 3.2.

The authors

[revised manuscript text omitted]

PAV 25/2/2020 11:46 — **Deleted:** 3…. The rainfall file has no … [1]

Auguste GIRES 27/2/2020 10:19 — **Deleted:** somea…periods with cumul… [2]

PAV 6/12/2019 16:10 — **Deleted:** m…rch (9 mm), 10 and …1… [3]

PAV 25/2/2020 11:45 — **Deleted:** 4

Auguste GIRES 27/2/2020 10:24 — **Deleted:** can

PAV 6/12/2019 13:15 — **Deleted:** ,…due to its …he granular … [4]

Auguste GIRES 27/2/2020 10:25 — **Deleted:** are

PAV 6/12/2019 15:18 — **Deleted:** two

Auguste GIRES 27/2/2020 10:25 — **Deleted:** data

PAV 6/12/2019 15:19 — **Deleted:** are

Auguste GIRES 27/2/2020 10:25 — **Deleted:** for the measure …n the …p… [5]

PAV 25/2/2020 09:02 — **Deleted:** operation

Auguste GIRES 27/2/2020 10:27 — **Deleted:** is

PAV 6/12/2019 12:55 — **Deleted:** Note …s already … [6]

PAV 25/2/2020 11:45 — **Deleted:** 4

[revised manuscript text omitted]

---

## Author Response (AR2)

Dear Editor,

We would like to thank the reviewer for this second review. His comments and remarks have been taken into account, as your remarks concerning the language.

The authors have made a reasonable attempt to address the reviewers' comments. Modifications to Figure 1, inclusion of a cross-sectional sketch for the roof system and inclusion of an individual storm event have all improved the manuscript.

However, questions around the moisture probe calibration remain. The authors have provided an alternative calibration equation (Equation 4) which gives vastly different values of theta compared with Equation 3. For example, for ka = 10, the calculated value for theta are 0.188 and 0.347 for Equations (3) and (4) respectively. Given this difference, and the acknowledgement that Topp's Equation may not be appropriate, it is unclear why the authors have chosen to use Equation 3 for the analysis presented in the paper.
In Section 3.3. it is suggested that this leads to rainfall storage in excess of the recorded rainfall, which does not lead to reader confidence in the data. The analysis should be done using the substrate specific calibration, Equation 4.

The use of Topp equation has been justified in Section 3 (coherency with previous literature). Note that the results obtained with the second relationship are also mentioned in Section 3.3. In both cases, among the 16 computed values, some of them overestimate the retained water. But for a large majority, the order of magnitude is consistent with the computed runoff coefficient, giving the reader confidence in the data.

In Figures 5 & 7, red and magenta lines are very difficult to distinguish between. A different colour combination should be used.

The colour chosen to draw the storage unit discharge has been changed (purple -> green).

In line 255-257 the authors state that they were particularly interested in understanding the influence of the slope on infiltration and evapotranspiration processes. I do not see any comments in the paper relating to this, and the data plotted in Figure 6 does not permit specific moisture content probes to be identified. Please comment on whether any systematic variations in moisture content or the rate of moisture content reduction in dry periods (evapotranspiration processes) were observed as a function of the probe position, the local slope and the relative location top/bottom of the wave.

A comment has been added in section 3.2 to mention that the lowest soil moisture values tend to refer to the upstream sensors (top of the wave), whereas the highest values tend to refer to the downstream ones (bottom of the wave). Sensor references have also been added in Figure 8 to illustrate this assertion.

The paper also requires careful editing for English. Some examples of repeated errors are in the use of 'performances' rather than 'performance', 'were comprised' rather than 'comprised', and 'coma' rather than 'comma'. The last line of the abstract is difficult to understand – maybe 'conducting' should be 'affecting'? Also line 64.

These errors have been corrected, those underlined in the marked manuscript too.

The authors

[revised manuscript text omitted]